# Preliminary Approaches to Cosmeceuticals Emulsions Based on N-ProlylPalmitoyl Tripeptide-56 Acetat-Bakuchiol Complex Intended to Combat Skin Oxidative Stress

**DOI:** 10.3390/ijms24087004

**Published:** 2023-04-10

**Authors:** Ana Simona Barna, Claudia Maxim, Adriana Trifan, Alexandra Cristina Blaga, Ramona Cimpoesu, Delia Turcov, Daniela Suteu

**Affiliations:** 1Department of Organic, Biochemical and Food Engineering, Faculty of Chemical Engineering and Environmental Protection “Cristofor Simionescu”, “Gheorghe Asachi” Technical University of Iasi, D. Mangeron Blvd., No. 73A, 700050 Iasi, Romania; 2Department of Pharmacognosy-Phytotherapy, Faculty of Pharmacy, “Grigore T. Popa” University of Medicine and Pharmacy, Universitatii Street, No. 16, 700115 Iasi, Romania; 3Department of Materials Science, Faculty of Materials Science and Engineering, “Gheorghe Asachi” Technical University of Iasi, D. Mangeron Blvd., No. 41, 700259 Iasi, Romania

**Keywords:** emulsions, bakuchiol, signal peptides, oxidative stress, skin protection

## Abstract

This study focuses on the development of a performant formulation for O/W dermato-cosmetic emulsions, which can be incorporated into novel dermato-cosmetic products or used as such. The O/W dermato-cosmetic emulsions contain an active complex based on a plant-derived monoterpene phenol, bakuchiol (BAK) and a signaling peptide named n-prolyl palmitoyl tripeptide-56 acetate (TPA). As a dispersed phase, we used a mix of vegetable oils, and as a continuous phase, *Rosa damascena* hydrosol was employed. Three emulsions containing different concentrations of the active complex were formulated (0.5% BAK + 0.5% TPA = E.1.1., 1% BAK + 1%TPA = E.1.2., 1% BAK + 2% TPA = E.1.3.). Stability testing was performed through sensory analysis, stability after centrifugation, conductivity and optical microscopy. A preliminary in vitro study regarding the diffusion ability of antioxidants through chicken skin was also undertaken. DPPH and ABTS assays were used to highlight the optimal concentration and combination in the formulation in terms of antioxidant properties and safety level of the active complex (BAK/TPA). Our results showed that the active complex used for preparing emulsions with BAK and TPA showed good antioxidant activity and is suitable for obtaining topical products with potential antiaging effects.

## 1. Introduction

Dermato-cosmetics (cosmeceuticals) are products at the interface between medicine and cosmetics, around which an industry already at the height of impressive performance has developed [1,2,3]. Amazingly, the more information, products and successful achievements appear, the more challenges that specialists from multiple research fields assume with determination. Dermato-cosmetic products must meet multiple demands: patients want pleasure in their use and multiple benefits from the same product, dermatologists recommend high safety and tolerability, along with safe therapeutic performance, and manufacturers prefer sources and technologies with optimal cost efficiency to profitability ratio. Thus, although in a field with a wide and high-quality offering, the competition between manufacturers, medical requirements and the increasingly sophisticated demands of consumers leave room for new improvements in dermato-cosmetics products [4,5,6].

With reference to active ingredients, the risks of modern life and their consequences make antioxidant ingredients valuable components of modern dermato-cosmetic products, especially when they bring, in addition to combating skin oxidative stress [7,8,9], other new benefits related to the appearance of the skin, through effects related to regeneration, repair, and soft imperfections such as pigmentation spots or fine wrinkles [9,10,11,12,13].

Cosmeceuticals emulsions are generally oil in water (O/W) emulsions, formulated by combining effective cosmetic ingredients with vegetable oils and active substances (antioxidants, antiaging agents, vitamins/provitamins) in amounts that offer a rich supply to the skin and demonstrate evident effectiveness [14,15]. The physiological processes can be activated more often by using certain components resulting in the effect of the skin’s natural regeneration. To avoid undesired effects, the amount of the excipients like preservatives, waxes, synthetic oils and other synthetic adjuvants must be reduced to a minimum level [1,4,16].

The modern approach of high-quality emulsions presenting long-term stability requires the addition of an emulsifier because they are thermodynamically unstable systems made up of at least one immiscible liquid that is intimately dispersed in another in the form of droplets [17,18]. The type of emulsifier included affects not only the final product’s functional characteristics and ease of emulsion formation but also its capacity to stabilize emulsions.

Modern emulsifying agents, such as amphiphilic polymers, are used in the cosmetics business to provide stability by absorbing at the oil/water interface and/or causing thickening. They lessen the amount of low-molecular-weight lubricants as well. Amphiphilic emulsifiers, which include small molecule surfactants, phospholipids, proteins, polysaccharides and other surface-active polymers, are organic substances with both hydrophilic and hydrophobic groups on the same molecule [18,19,20,21].

Nature remains a valuable provider of active ingredients, with advantages related to safety in use and accessibility as well as performance benefits like efficacy due to a wide range of therapeutic actions and biotechnological advantages [5,6,22,23,24].

Vegetable oils are widely used in cosmetic formulations due to their emollient properties, protecting the skin barrier and restoring cutaneous homeostasis. They replace mineral oils, given customers’ growing interest in herbal products with few or any side effects, allergy issues or occlusive effects [25].

Vegetable oils have beneficial effects on and in the skin as a result of the C-chain distribution in triglycerides and their accompanying substances. Due to the chemical composition of fatty acids, glycerol and triglycerides, vegetable oils strengthen the lipid barrier layer, stop water loss through the skin, soften and smooth the stratum corneum and lessen skin inflammation. In contrast to vegetal oils, mineral oils should not be used on the skin because they can have occlusive effects [26,27].

A good dermato-cosmetic O/W emulsion should include vegetable oils in an amount of at least 10% in addition to antioxidants, vitamins and moisturizing agents. In O/W emulsions, the lipid phase usually has a proportion of 10–20%. Several vegetable oils (*Euterpe oleracea oil*—Açai oil, *Oenothera biennis oil*—Evening Primrose, *Punica granatum seed oil*—Pomegranate seed oil) were selected for cosmetic formulation as a lipid phase. The selection was based on their composition and its correlation with antioxidant activity.

The focus of antiaging research is on substances that inhibit harmful cellular oxidation using radical scavengers that absorb and neutralize free radical species. Antioxidants are substances that slow down premature skin aging [10,12,13].

Vegetable oils rich in vitamins are also beneficial for skin regeneration due to their active fatty acids. The second key area of research is identifying substances that can increase cell production [28,29,30,31,32].

Bakuchiol (BAK) [1-(4-hydroxyphenyl)-3,7-dimethyl-3-vinyl-1,6-octadiene] is a natural meroterpenoid abundant in the plant *Psoralea corylifolia (Fabaceae* family). The additional plant species that contain bakuchiol are *Prosopis glandulosa*, *Otholobium pubescens*, *Pimelea drupacea*, *Ulmus davidiana*, *Piper longum*, *Aerva sanguinolenta*, *Psoralidium tenuiorum*, *Bridelia retusa*, *Elaeagnus bockii*, *Spiraea formosana* and *Nepeta angustifolia*. *P. corylifolia* is the only one of these that can currently be used to produce bakuchiol naturally on a larger scale.

The most prevalent substance in this species, bakuchiol, accounts for 6.24% of the weight of the dried seed. *P. corylifolia* is a source of several other meroterpenoids besides bakuchiol with antioxidant activity. Bakuchiol (BAK) has potential antiaging, anti-inflammatory, and antibacterial properties and is considered an essential ingredient in skincare and a better retinoid alternative [33,34,35], causing less or no dryness, redness, and irritation.

Based on experience and knowledge of other related molecules, such as resveratrol, it is also possible to predict a bakuchiol molecule’s bioactivity from its molecular entities (pharmacophores). Because they both contain a 4-hydroxystyryl moiety, bakuchiol and resveratrol have structural similarities. Additionally, the pyrone, chromene and quinazoline derivatives linked by the styryl moiety reported strong biological properties. Due to the similar pharmacophore, it is possible that the bakuchiol moiety will exhibit properties that are similar to those of resveratrol and other related molecules.

Several peptide variants represent another research direction and have been used as antiaging cream ingredients since the late 1990s. The peptides function as versatile messengers in the skin and act as signaling molecules that initiate cell metabolism. This translates to better skin cell formation, better skin regeneration and increased collagen synthesis. Skin and contours become firmer as a result. Other peptides, on the other hand, have the ability to relax the facial muscles, which can tighten mild facial lines. [36,37,38].

In this study, in synergy with bakuchiol (BAK), a small signal peptide called n-prolyl palmitoyl tripeptide-56 acetate (TPA) was used, with the potential to increase the production of collagen, glucosaminoglycan, elastin and fibronectin at the gene and/or protein levels.

The dermato-cosmetic formulation in this study presents a combination of self-assembled amphiphilic polymers, PEG-free, made from entirely renewable resources with emulsifying capability in cosmeceuticals formulation to develop low-viscous emulsions [39]. Besides remarkable stabilization ability, the present association of high-performant emulsifying polymers (polyglyceryl 6 stearate and polyglyceryl 6 behenate) is also considered to have excellent moisturization qualities.

The purpose of this study is to study emulsions (which can be incorporated into new dermato-cosmetic products or used as a finished product) based on bakuchiol (BAK), a small signal peptide called n-prolyl palmitoyl tripeptide-56 acetate (TPA) and an association of high-performant emulsifying polymers (polyglyceryl 6 stearate and polyglyceryl 6 behenate) in order to determine their stability and dispersion of active ingredients, which confers quality, safety and efficacy of the final product. In addition, in vitro studies bring valuable preliminary information regarding the diffusion ability of the antioxidants in the product (quantified in the form of total polyphenols content) through the chicken membrane using Franz cell tests.

## 2. Results

Several emulsions were formulated using as active ingredients an active complex containing bakuchiol (BAK) and a signal peptide, n-prolyl palmitoyl tripeptide-56 acetate (TPA), with different concentrations (0.5% BAK + 0.5% TPA = E.1.1., 1% BAK + 1%TPA = E.1.2. and 1% BAK + 2% TPA = E.1.3.).

One type of emulsifier may be sufficient to create an emulsion in some circumstances. Instead of using individual emulsifiers, combinations of emulsifiers can be used in many applications to improve the formation, stability and functional attributes of emulsions [18,39,40,41].

The stability of the prepared emulsions was assessed throughout time as well as potential microbial contamination during storage or usage of dermato-cosmetic preparation [37,39,40,42,43,44,45,46,47,48,49,50,51,52,53].

### 2.1. Stability Evaluation

#### 2.1.1. pH Measurements

The pH values of the emulsions, in the range of 4.5 to 6 (Figure 1a), were deemed adequate to prevent the risk of irritation upon application to the skin.

#### 2.1.2. Conductivity Measurements

The release of the electrolyte initially contained in the internal water phase was investigated using conductometric analysis [54,55]. The results of conductivity measurements ranged from 0.36 to 0.52 mS (Figure 1b).

#### 2.1.3. Phase Separation

*Centrifugation test*. The physical stability of O/W emulsions was examined by the Centrifugation test to demonstrate their resistance to external factors such as centrifugal force [43]. After extensive centrifugation, there was no evidence of phase separation in any of the samples (Table 1)

*Vibration test*. Vibration during transportation can impact formulation stability by inducing emulsion phase separation, suspension solidification and viscosity alterations (ANVISA, 2005). The cosmetic compositions passed the vibration test without revealing any phase separation, proving their physical stability (Table 1).

### 2.2. Antioxidant Activity

The first phase of the investigations on the antioxidant activity of the formulated emulsions intended to determine the total amount of compounds with antioxidant action, total phenolic content (TPC) [56]. The studies continued with the assessment of the antioxidant activity displayed by the emulsions. Table 2 presents the corresponding data.

### 2.3. Microbiological Control

The microbiological analysis of a cosmetic product is necessary to guarantee the conformity of that product when it is placed on the market with regard to safety during use by the final consumer. The products are usually analyzed in accordance with ISO 18415:2007 standard, approved by the European Union Regulation on cosmetic products, EC Regulation 1223/2009. Microbiological control ensures the emulsion stability over time and verifies proper emulsion preservation. Cosmetic products are not sterile products; the standards accept that the total number of viable aerobic mesophilic microorganism should not exceed 10^3^ CFU/g or 10^3^ CFU/mL of product, but *Pseudomonas aeruginosa* and *Staphylococcus aureus* should not be detectable in the product. A consumer will thereby be assured that any cosmetic product is within the established microbiological limits and will remain so until the product is consumed. The following are typical tests to assess a cosmetic product’s microbiological quality: mesophilic aerobic microorganism count, most probable number (MPN) applied for coliforms, yeast and mold count, and presence of *Staphylococcus aureus* and *Pseudomonas aeruginosa*.

The main objective of cosmetic microbiological testing is to guarantee product quality and purity. Anywhere there are favorable conditions (appropriate temperatures, plenty of nutrients, and moisture), microorganisms will proliferate. Numerous cosmetics, especially emulsion-type formulations, serve as favorable development environments for bacteria and fungi (yeasts and molds). Therefore, it is crucial to find ways to stop their growth in order to stop the product from degrading and to guarantee the user’s safety while using it. Product spoilage over time is frequently indicated by the emergence of unpleasant odors, changes in color or texture, and phase separation. The microbiological control of cosmetics is subject to required legal and regulatory standards in the European Union (EU), the United States and Japan. To guarantee the formulas’ excellent performance and safety and to confirm that these qualities endure over time, quality control of cosmetic products is essential [57,58,59,60].

The results obtained for the proposed emulsions are presented in Table 3.

### 2.4. Analysis of Homogeneity and Stability of the Emulsions

#### 2.4.1. Scanning Electron Microscopy and Energy Dispersive X-ray Spectroscopy

The surface morphologies and compositions were analyzed through scanning electron microscopy for the characterization of homogeneity and stability of the emulsion without active substances (E 1.0.) and emulsion with 1% Bak + 2% TPA (E.1.3.). Furthermore, micrographs and elemental distributions are presented for each emulsion sample.

*Scanning electron microscopy* was used to study the base of emulsion (E. 1.0.) and the base with active ingredients—emulsion E.1.3. Images obtained at 50, 100, 500 and 1000× magnifications are shown in Figure 2.

Also, the chemical composition of the emulsion without and with actives was characterized using energy-dispersive X-ray spectroscopy (EDX). The determinations were performed in three areas on the surface of the material (1 mm^2^) and an average was used. The spectra analysis of the emulsion without actives (Figure 3a) identified C, O and N at different X-ray energies, and the spectra of the emulsion with actives (Figure 3b) revealed the same composition, uniform with slight changes as a consequence of incorporating the assets in the specified concentrations, of 1% BAK and 2% TPA.

#### 2.4.2. Microscopic Image Analysis

Table 4 illustrates the obtained optical microscopy images of the studied emulsions.

### 2.5. Preliminary Results of In Vitro Diffusion Study

The results of this study were expressed in three ways that have different meanings, as follows: (i) polyphenols released in 5 mL from the receptor chamber, expressed in TPC, μg/mL as a function of time (Figure 4); (ii) the release rate of polyphenols, expressed in TPC μg/mL/t (Figure 4); and (iii) the efficiency of penetration of polyphenols through the membrane (Figure 5).

## 3. Discussion

### 3.1. Stability Evaluation

#### 3.1.1. pH Measurements

The dermato-cosmetic emulsions must correspond to the pH of the skin to be well tolerated. Race, skin color, sun exposure, infections and various products used on the skin can all affect pH levels. Since the normal skin pH is between 4.5 and 6, the pH values of the emulsions, which varied from 4.5 to 6, were considered acceptable to prevent the risk of irritation when applied to the skin. It is essential for preserving the integrity of the epidermis’ lipid barrier and stratum corneum that the formulated dermato-cosmetic emulsions are in accord with the skin pH [61].

The pH stability of the emulsions is a crucial aspect, as changes in pH values indicate either the occurrence of side chemical reactions in the formulation or the decrease of preservative effectiveness at a certain pH level and the appearance of microbial contamination. In the current work, pH testing up to 30 days of storage was conducted for all formulated emulsions. In all cases, the pH values showed values between 4.65–5.54. Compared to the blank formulation (E.1.0.—emulsion without active substances) pH = 4.65, the emulsions with an active biologic complex showed a slightly higher pH (4.8–5.54), which may be explained by the incorporation of the bakuchiol and signal peptide (n-prolyl palmitoyl tripeptide-56 acetate) in the formula (Figure 1a).

Therefore, from these measurements, it can be concluded that the studied emulsions showed steady pH values during storage. Considering that skin pH ranges between 4 and 6, the O/W emulsions with a biologically active complex containing bakuchiol and n-prolyl palmitoyl tripeptide-56 acetate are compatible with the skin and suitable for cosmetic dermal applications.

#### 3.1.2. Conductivity Measurements

The behavior of the green complex of active substances (BAK+TPA) included in the inner phase of the primary emulsion was also studied using this method. The effects will become less long-lasting the more electrolyte is released, and the active substance is free to move in the external aqueous phase. There are some factors that might influence the conductivity values while the product is being stored. The diffusion of the active ingredient, the coalescing of the internal and aqueous phases or the destruction of the oil film by osmotic pressure and internal aqueous phase leaking will result in a conductivity increase [43]. The results of the measurements (Figure 1b) show that the emulsions under examination are stable throughout time.

#### 3.1.3. Phase Separation

The dermato-cosmetic compositions passed the *centrifugation* and *vibration tests* without revealing any phase separation, proving their physical stability during transport and handling conditions (Table 1).

### 3.2. Antioxidant Activity

Our results (Table 2) showed an increase in the antioxidant power of the emulsions as compared to the base. It is also observed that the antioxidant capacity can be correlated with the percentage of bakuchiol and signal peptide (TPA) and also with the total phenolic content. Bakuchiol has already been highlighted as a quenching agent in several in vitro studies. Blumke et al. showed that at a concentration of 100 µM, bakuchiol scavenged DPPH radicals to a greater extent than retinol [62]. At a similar concentration, the meroterpene exhibited 84.6% ABTS radical scavenging activity, a value comparable with that of the positive control resveratrol [63]. Oxidative stress on skin cells, either from internal metabolic processes or external environmental stressors, is known to contribute significantly to cutaneous aging. The antioxidant properties of bakuchiol were outlined as an important mechanism against photoaging. Thus, its involvement in several antioxidant processes (e.g., oxygen radicals scavenging, inhibition of lipid peroxidation, activation of nuclear factor erythroid 2-related factor 2, a transcription factor involved in the cell resistance to oxidative stress) [64,65,66] support bakuchiol’s efficacy in topical skincare treatments against photoaging, as already shown by some clinical investigations [35,62].

### 3.3. Microbiological Control

The obtained results (Table 3) show that each formulation has very low or zero contamination, all values being inferior to the values accepted by the standards (10^3^ CFU/g colony-forming unit). The added compounds (BAK and TPA) have increased the microbiological stability of the proposed formulations, as the number of CFUs decreases to 0 with the increase of BAK and TPA from 0.5 to 1.5%. The absence of pathogenic strains in the base without any supplements showed that the formulations are safe to use, even if the total viable microbiological count is 80 CFU/g. Taking into account possible contamination, it is crucial to find ways to stop microbial growth in order to prevent product degradation and to guarantee the user’s safety while using it. Bakuchiol is a meroterpene with a partial terpenoid structure that possess antimicrobial activity in addition to its other effects (anti-inflammatory, antioxidative or antiosteoporotic) [67].

### 3.4. Analysis of Homogeneity and Stability of the Emulsions

#### 3.4.1. Scanning Electron Microscopy and Energy Dispersive X-ray Spectroscopy

The relevant information from these two methods of analysis (SEM and EDX) in Figure 2 and Figure 3 underlines the fact that the analyzed emulsions (with or without actives) are systems characterized by homogeneity and stability of the phases during the storage period.

#### 3.4.2. Microscopic Image Analysis

As can be observed in Table 4, the emulsions display a stable morphology after 30 days of storage under normal temperature conditions. Optical images show that all emulsions are polydisperse and the molecules of the dispersed phase are distributed as a compact mass of small globules. This fact also demonstrates phase homogeneity during the storage period.

### 3.5. Preliminary Results of In Vitro Diffusion Study

As depicted in Figure 4, the release rate is very fast and obvious at the beginning of the release process, when the concentration gradient is the highest, and gradually decreases to a much lower value at an interval of 6 h.

The low release efficiency (E%) (Figure 5), which is also desirable in the case of these types of emulsions, is explained by the fact that the active ingredients are released primarily in the superficial layers located toward the chicken membrane. It should also be considered that the system as a whole has relatively limited dynamics, especially in the donor compartment, while the relative viscosity of the formulations contributes to this behavior.

In vitro test results complete our earlier investigations and demonstrate that this emulsion can be incorporated into novel dermato-cosmetic formulations (or be used by itself) with the goal of reducing skin oxidative stress. The objective of emphasizing the diffusion system’s functionality in the context of using a new emulsion has been accomplished, and this study serves as an important first step for future research.

## 4. Materials and Methods

### 4.1. Materials

#### Active Compounds

A biologically active complex containing a plant-derived meroterpene phenol, bakuchiol (BAK) [1-(4-hydroxyphenyl)-3,7-dimethyl-3-vinyl-1,6-octadiene] (Figure 6) and a peptide as a n-prolyl palmitoyl tripeptide-56 acetate (TPA). BAK (≥98% purity) was purchased from Elemental SRL (Oradea, Romania). It is an alternative to retinoids. TPA (MatrIxyl^®^ Morphomics) is a small semnal peptide purchased from Glamourcosmetics (Italy) in water, pentylene glycol and caprylyl glycol support.

A mix of two self-assembled amphiphilic copolymers, polyglyceryl 6 stearate (85–98%) (C_36_H_72_O_14_) and polyglyceryl 6 behenate (2–15%) (C_40_H_82_O_15_), was used as an emulsifier for the O/W emulsions. This mix of emulsifiers was supplied as Liqua^®^ commercial product from Elemental SRL (Romania). HLB (the value for this emulsifier is 13; a coemulsifier—glyceryl stearate—was used in conjunction with the emulsifiers to thicken and stabilize emulsions; Glyceryl stearate (Glycemul^®^) was purchased from Elemental SRL (Oradea, Romania). Xanthan gum is an effective thickening agent and stabilizer of emulsions. It was purchased from Elemental SRL (Romania).

A mix of vegetable oils was used as the dispersed phase, *Euterpe oleracea oil*, *Oenothera biennis oil* and *Punica granatum seed oil*, and as the continuous phase, *Rosa damascena* hydrosol was used. The dispersed phase and the continuous phase were supplied from Aromazone (Avignon, France).

Antimicrobial conservation of emulsions was achieved by the use of a mix of substances: benzyl alcohol, salicylic acid, glycerin, sorbic acid (Cosgard^®^), purchased from Aromazone (France). All chemical reagents used for analytical analysis were of analytical purity (p.a.).

### 4.2. Formulation of O/W Emulsions

The formulation of the dermato-cosmetic oil/water (O/W) emulsions was conducted using vegetal ingredients. Table 5 summarizes the ingredients used for the studied formulations. The continuous phase (aqueous phase) and the dispersed phase (oil phase) were heated up to 75 °C. The oil phase was then added dropwise to the water phase in the O/W ratio of 23/77 under continuous mechanical stirring using a rotor–stator homogenization (ESGE Zauberstab M 160 G Gourmet) operating at 15,000 rpm.

The resulting emulsion was cooled down to 40 °C and the active biologic substances in the established concentration (0.5% BAK + 0.5% TPA = E.1.1., 1% BAK + 1%TPA = E.1.2. and 1% BAK + 2% TPA = E.1.3.) and the cosmetic preservative was added under stirring.

Finally, a blank emulsion without the addition of any bioactive (E.1.0.) was also prepared according to the procedure described above. For further studies, 30 g samples were weighed and then packed in airless containers kept in optimal conditions (temperature of 20 ± 2 °C).

### 4.3. Methods

#### 4.3.1. Stability Evaluation

The stability of the products was assessed based on “Cosmetics Europe Guidelines on Stability Testing of Cosmetic Products 2004” [38].

O/W emulsions can break down over time by different processes such as creaming, flocculation, coalescence and phase inversion. The quality of a new product can be assessed by performing a set of analyses that also aim at the stability of the resulting emulsion. To predict the stability over time of dermato-cosmetic emulsions, several analyses were performed under specific conditions [42,68,69], including pH determination, phase separation under centrifugal force, conductivity determination, homogeneity assessment and microbiological control.

All samples used for analysis were kept at room temperature. Depending on the type of analysis, stability assessments (pH, conductivity and centrifugation tests) were conducted at various points during the preparation phase. The following time intervals were considered in these investigations: the moment immediately after preparation, seven days later, and one month following preparation and storage under standard circumstances (temperature of 25 °C, in airless containers composed of material that blocks the entrance of light rays) [37,68,70].

##### pH Measurements

The pH determination of the investigated samples was performed according to the protocol presented in our previous work [43]. For the determination of the pH values of dermato-cosmetic emulsions, a digital pH Meter (Hanna Instrument, Mauritius) was used. The pH measurements were also taken for the samples at 24 h and 30 days. The pH values of emulsions, determined in duplicate, were measured at room temperature (20 ± 2 °C), and the average values were calculated.

##### Conductivity Measurements

The specific conductivity measurements of undiluted emulsion samples were done using a portable Hanna Instrument type conductometer (Nusfalau, Romania). Conductivity tests were performed for emulsion formulation at 24 h and 30 days. The determination of emulsions conductivity was done at room temperature (20 ± 2 °C). All the measurements were performed in duplicate.

##### Phase Separation

The phase separation analysis of each formulation was performed under the action of centrifugal and vibrational force and following our previous protocol [43].

Centrifugation test. To perform the centrifugation test, 5 g of each formulation were submitted to 3 cycles of 3000 rpm for 30 min at 25 °C. At the end of each cycle, the samples were checked to see if any changes of the emulsion texture occurred. Centrifugation assay was carried out using a Centrifuge XC-Spinplus (Shanghai, China).

Mechanical Vibration test. This test investigates the stability of the emulsions under mechanical vibration movement, which may cause instability of formulation detected as phase separation. Briefly, 5 g of sample were subjected to vibration on a vortex shaker-a Multi Speed Vortex MSV-3500 (Grant Instruments Ltd., Cambs, UK), for 30 s. The test was carried out in accordance with the protocol given in our prior work [43].

#### 4.3.2. Assessment of Total Phenolic Contents in Emulsions

The analyses were performed using the dermato-cosmetic emulsion extracts thus obtained: 0.5 g of individual emulsion was extracted with 10 mL ethanol using a magnetic stirrer (30 min at room temperature), followed by filtration (0.22 µm pore diameter) (a similar methodology used in our previous study [44]).

The total phenolic content was determined following previously described methods [45,46,47]. Briefly, 50 µL of sample were mixed with 100 µL Folin–Ciocalteu reagent and strongly mixed. After 3 min, 75 µL of 1% Na_2_CO_3_ solution were added and the mixture was incubated for 2 h at room temperature in the dark. After reaching this time, the absorbance was read at 760 nm against a blank solution prepared from the 50 µL sample with 100 µL Folin–Ciocalteu reagent without sodium carbonate, employing a SpectroStar Nano Microplate Reader (BMG Labtech, Ortenberg, Germany). The total phenolic content was expressed as milligrams of gallic acid equivalents (milligrams GAE/g emulsion). The results were calculated as mean ± standard deviation (SD) of three determinations.

#### 4.3.3. Determination of Antioxidant Activity of Emulsions

##### 2,2-Diphenyl-1-picrylhydrazyl Radical Scavenging Assay (DPPH Method)

The experiment was carried out using a slightly modified version of a procedure that was previously described [45,46,47]. Thus, 50 µL of emulsion extract was added to 150 µL of 2,2-diphenyl-1-picrylhydrazyl (DPPH) 0.004% methanol solution. Following 30 min incubation in the dark at room temperature, the absorbance was read at 517 nm. DPPH radical scavenging activity was expressed as milligrams of Trolox equivalents (mg TE/g emulsion). The results were calculated as mean ± standard deviation (SD) of three determinations.

##### 2,2′-Azino-bis(3-ethylbenzothiazoline) 6-sulfonic acid) Radical Scavenging Assay (ABTS Method)

The experiment was conducted based on the adaptation of the procedure previously described [46,48,49]. ABTS•^+^ was generated by mixing 7 mM 2,2′-azino-bis(3-ethylbenzothiazoline) 6-sulfonic acid (ABTS) solution with 2.45 mM potassium persulfate (1:1, *v/v*). The mixture was allowed to stand for 12–16 min in the dark at room temperature. At the beginning of the assay, ABTS solution was diluted with methanol to reach an absorbance of 0.700 ± 0.02 at 734 nm. Then, 30 µL of emulsion extract was added to 200 µL ABTS solution and vigorously mixed. After 30 min incubation at room temperature, the absorbance was read at 734 nm. The ABTS radical scavenging activity was expressed as milligrams of Trolox equivalents (mg TE/g emulsion). The results were calculated as mean ± standard deviation (SD) of three determinations.

#### 4.3.4. Microbiological Control

The microbiological analysis was carried out for all aerobic mesophilic microorganisms (total aerobic microbial count and total yeast and mold count), following an amended protocol from ISO 18415:2007 [49]. The analysis was realized at 48 h following the cream’s preparation and packaging. Every operation for microbiological control was carried out in a laminar flow hood (Steril Helios MI2754b, Milano, Italy). An aqueous solution containing 1% HCl and 70% ethanol was used to clean the workspace and the containers and all samples were analyzed in triplicate, and the medium sterility control was also performed. Each sample, weighing a total of 1 g, was taken aseptically and added into a sterile test tube containing 5 glass beads (5 mm diameter), 1 mL of sterile Tween 80 and MLB—Modified Letheen Broth (approx. 8 mL) for the 1:10 dilution and mixed for homogeneity. 0.1 mL of this sample was inoculated on a sterile MLA—Modified Letheen Agar Petri dish (90 mm) using the streak plate technique and the plates were incubated for 1 to 5 days at 30 °C using a Biobase Incubator BOV-V35F (Shandong, China). The resulting colonies were counted using a colony counter (Scienceware ULB-100, Shandong, China) to determine the total number of viable colonies developed on each plate. The detection parameter was provided as present or absent/g, while the overall microbial count was reported as CFU/g, accounting for the dilution factor. To examine pathogenic strains of *Staphylococcus aureus*, *Escherichia coli*, *Enterobacter aerogenes*, *Pseudomonas aeruginosa* or *Salmonella*, microorganisms isolated from the Petri dish were grown on selective media: Mannitol Salt Agar, MacConkey Agar, Eosin Methylene Blue, Cetrimide Agar, and Brilliant Green Phenol Red Agar.

#### 4.3.5. Analysis of Homogeneity and Stability of the Emulsions

##### Microscopic Image Analysis

Microscopic image analysis was used to study the morphology and homogeneity of the emulsion. For this purpose, emulsion samples were observed after 30 days of storage under normal conditions. The images were monitored and captured by a binocular microscope Optika B-159 (OPTIKA S.r.l., Ponteranica (BG), Bergamo, Italy), magnification—1000×.

##### Physical-Chemical Characterization of Emulsions

For the analysis through *electronic microscopy* (*SEM*) and through *energy dispersive spectroscopy* (*EDX*), the samples were maintained in a vacuum for 24 h. The samples were then analyzed by primary beam at 8 kV power supply voltage. Images were obtained with a Secondary Electrons (SEs) detector (WD 15.5 mm, 30 kV, HV) from a scanning electron microscope (SEM): Vega Tescan-LMHII (Tescan Orsay Holding, Brno-Kohoutovice, Czech Republic). An automatic mode was used for determinations of chemical composition with Energy Dispersive Spectroscopy equipment, Bruker Nano GmbH Berlin, Germany. For mapping the distribution of the elements, Esprit 2.2 software was used in automatic mode.

#### 4.3.6. In Vitro Diffusion Methodology

Preliminary permeation studies were developed using the diffusion methodology based on Franz cell in order to investigate as appropriate the application of bakuchiol-based emulsions by the cutaneous route and to establish the required data for extending this study to real skin (human skin) in order to investigate real-like behavior. The cell was equipped with a membrane represented by fresh chicken skin (selected because it behaves similarly to human skin) with a thickness of 0.85 mm, which was previously prepared by degreasing. For this purpose, a commercial prepeeling solution based mainly on denatured alcohol and salicylic acid (Mesoestetic, Spain) was used, followed by washing with double-distilled water. From the skin thus prepared and dried with filter paper, a cropped piece ensured a permeation diameter of 1 cm^2^.

The working protocol was adapted according to our previous study [50] and information from the literature [51,52]. A precise amount of emulsion E.1.3. (selected for the highest amount of polyphenols) was added to the donor compartment and approximately 5 mL of isotonic solution with pH = 7 (which simulates the pH of human skin layers) [53] was added to the receiver compartment.

The Franz cell was continuously maintained on a magnetic stirrer at 37 °C temperature. Two hundred μL emulsion E 1.3. samples were taken out of the receptor chamber at predetermined intervals (ranging from 10 min to 6 h), and the removed volume was replaced with an exact equivalent volume of isotonic solution to maintain constant the compartment’s volume. In the samples taken, the amount of total polyphenols transferred through the chicken skin membrane of the Franz cell was determined.

## 5. Conclusions

In the present study, three types of oil/water dermatocosmetic emulsions containing a synergistic bioactive complex based on bakuchiol and n-prolyl palmitoyl tripeptide-56 acetate were prepared. As a dispersed phase, we used a mix of vegetable oil, and as a continuous phase, we used *Rosa damascena* hydrosol. Polyglyceryl 6 stearate and polyglyceryl 6 behenate were used as emulsifiers because they are amphiphilic chemical compounds based on renewable raw materials.

The formulated emulsions were found to have good stability, good compatibility of components over time and resistance to microbiological contamination. Also, the results from in vitro tests wrap up the preliminary research and show that the emulsions could be used *per se* or incorporated in novel dermato-cosmetic formulations intended to combat skin oxidative stress.

The chemical engineering contribution of this research conclusion is a required step in any new dermato-cosmetic formulation. Skin homeostasis, with specific mechanisms and molecular balances, represents a critical standard for a high-quality product; therefore, any research in this respect provides valuable input for a successful new product. Future research will consider those formulations that, according to certain protocols and following future in vivo studies, present high stability and impeccable behavior.

## Figures and Tables

**Figure 1 ijms-24-07004-f001:**
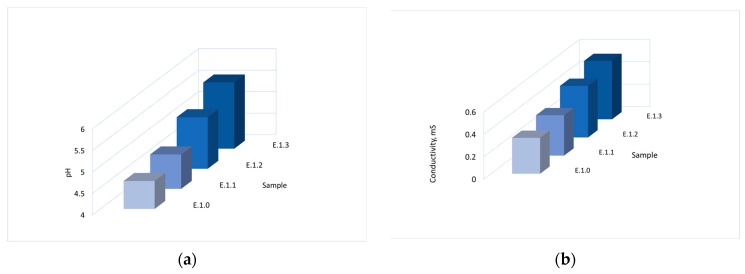
pH (**a**) and conductivity (**b**) of the studied emulsions.

**Figure 2 ijms-24-07004-f002:**
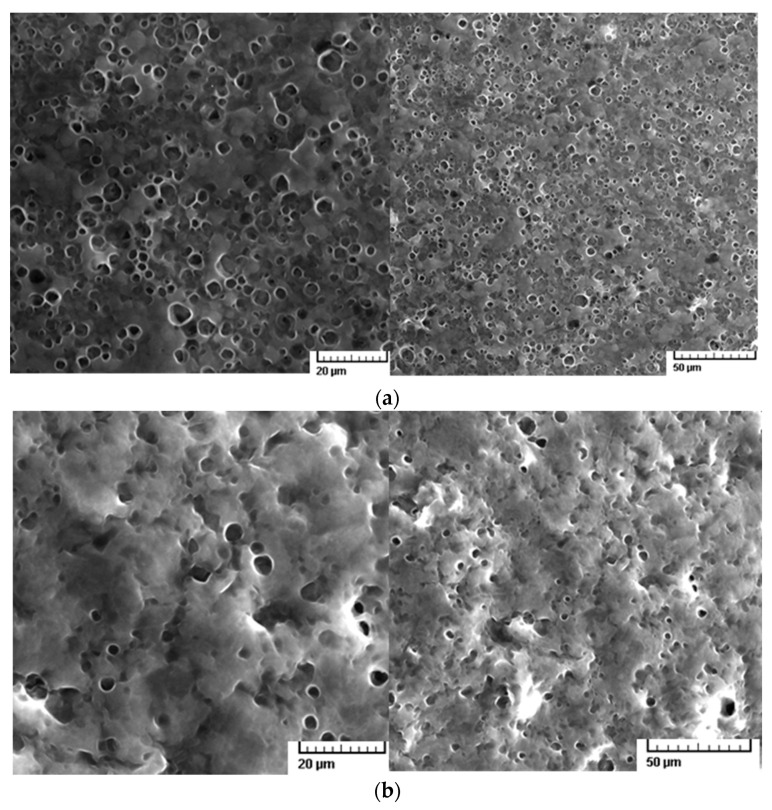
Scanning electron microscopy (SEM) of the emulsion E.1.0. (**a**) and emulsion E.1.3. (**b**).

**Figure 3 ijms-24-07004-f003:**
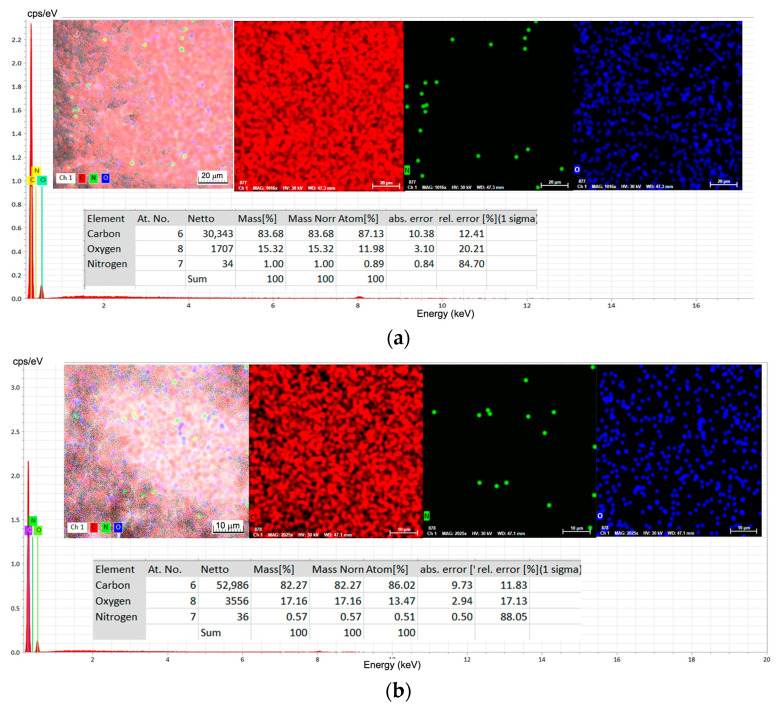
Energy-dispersive X ray (EDX) spectra of the of the emulsion E.1.0.—without active (**a**) and emulsion E.1.3.—with active (**b**).

**Figure 4 ijms-24-07004-f004:**
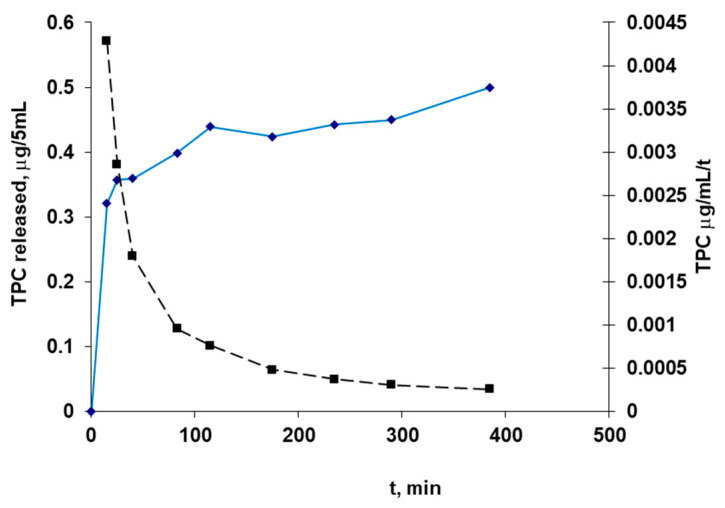
The amount of polyphenols (TPC) released in the receptor compartment (5 mL) by the E 1.3. emulsion: TPC μg/5 mL (blue line) and the rate of release of polyphenols, expressed in TPC μg/mL/t (black line), depending on time.

**Figure 5 ijms-24-07004-f005:**
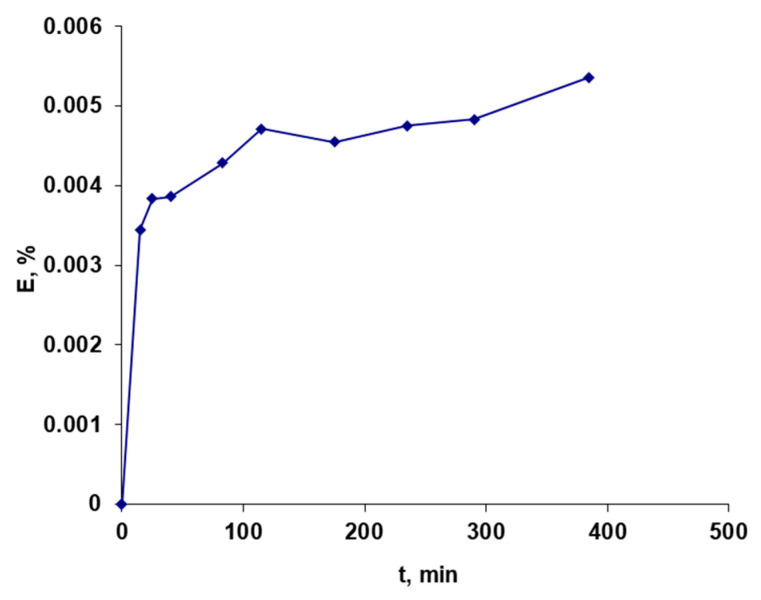
Efficiency of permeation of polyphenols from E 1.3. emulsion through chicken skin.

**Figure 6 ijms-24-07004-f006:**
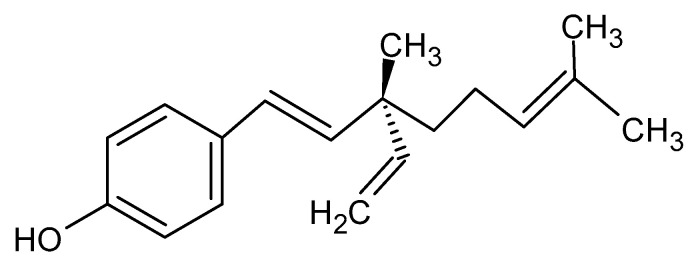
Chemical structure of bakuchiol.

**Table 1 ijms-24-07004-t001:** The results of the centrifugation and vortex stability tests.

**Dermato-Cosmetic Emulsion Samples**			
E.1.0. without active substances	E.1.1. 0.5% Bak 0.5% TPA	E.1.2. 1% Bak + 1% TPA	E.1.3. 1% Bak + 2% TPA
**Dermato-cosmetic emulsion stability after centrifugation test**			
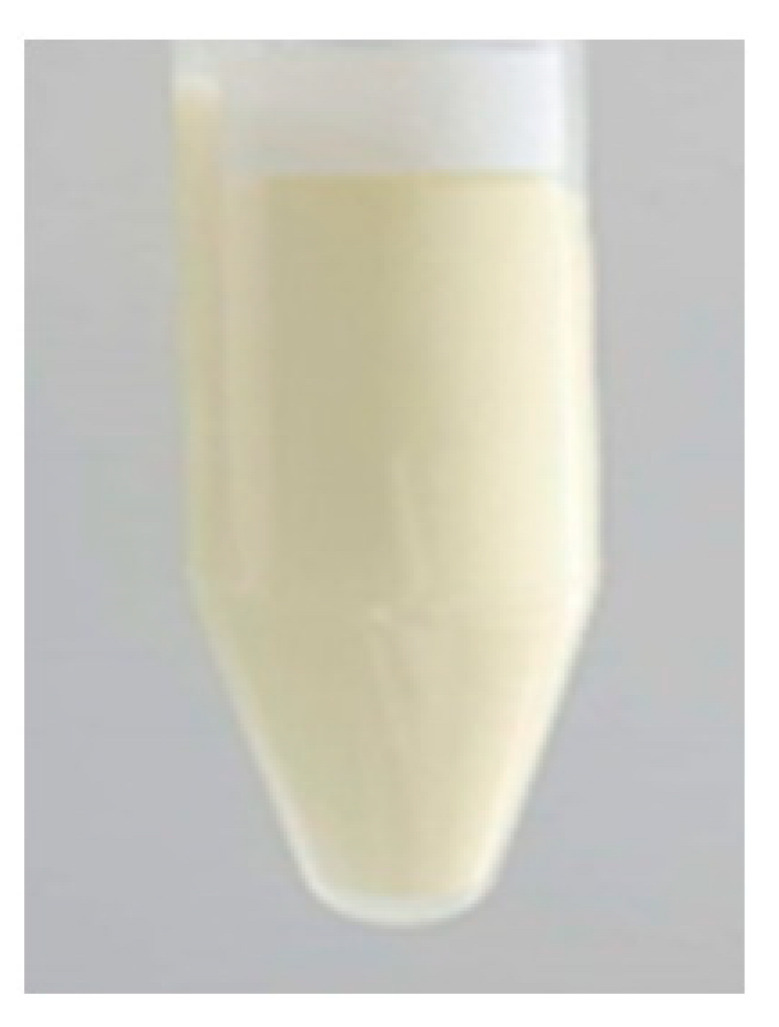	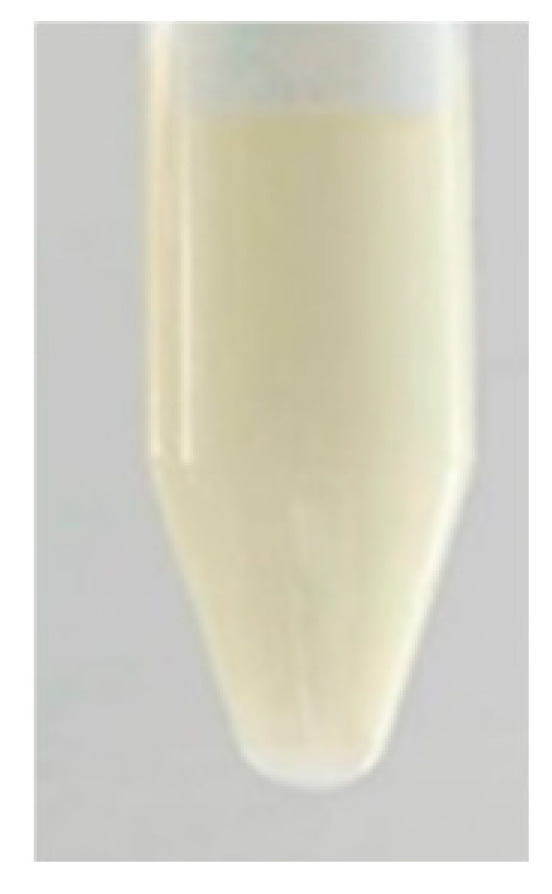	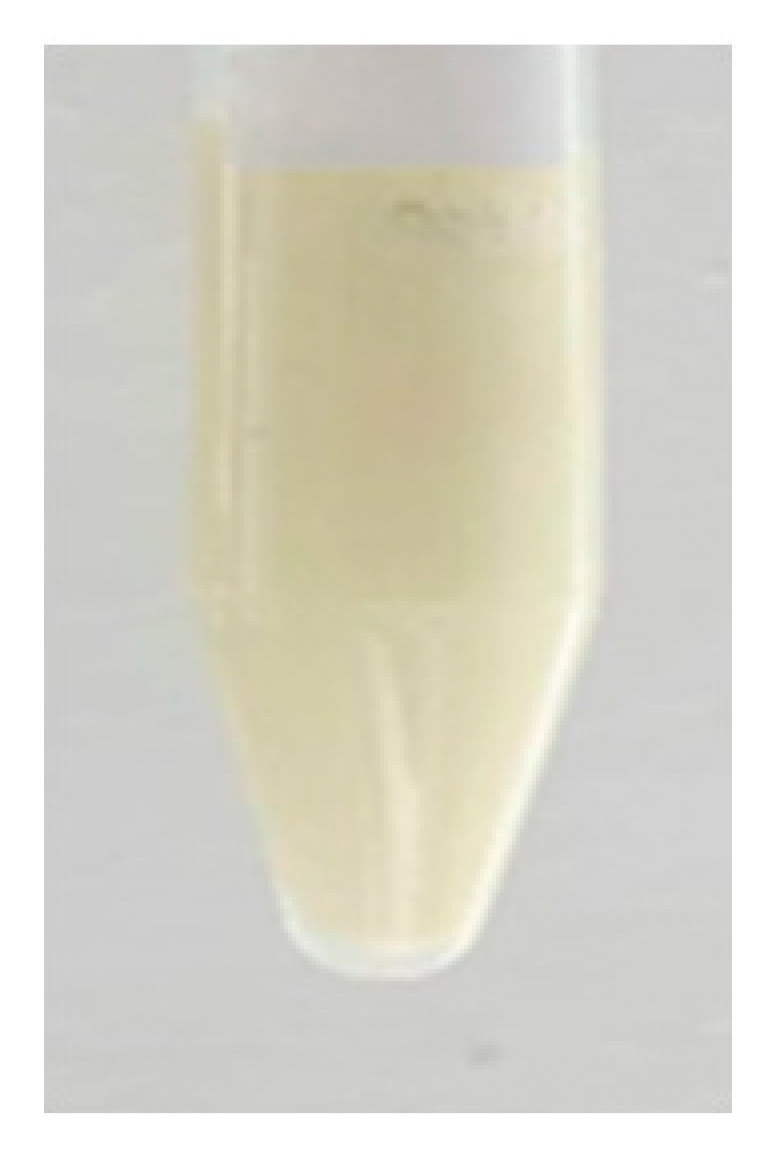	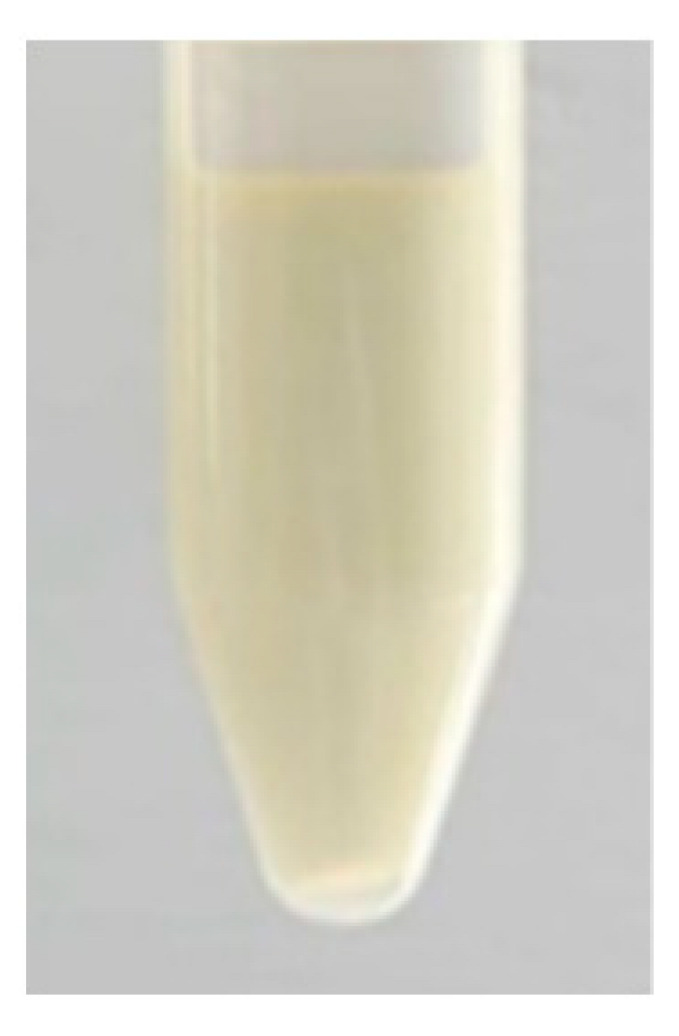
**Dermato-cosmetic emulsion stability after vortex test**			
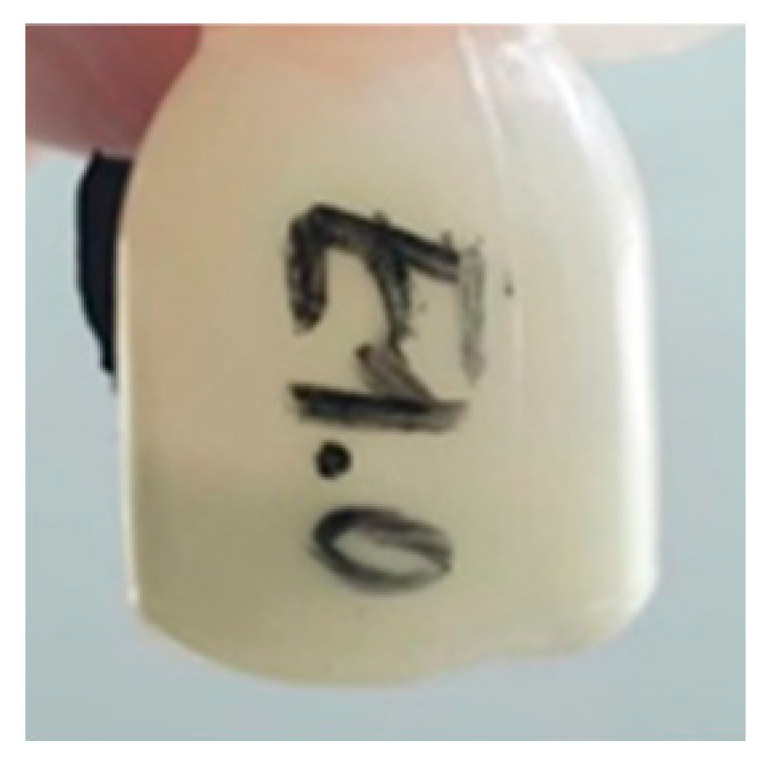	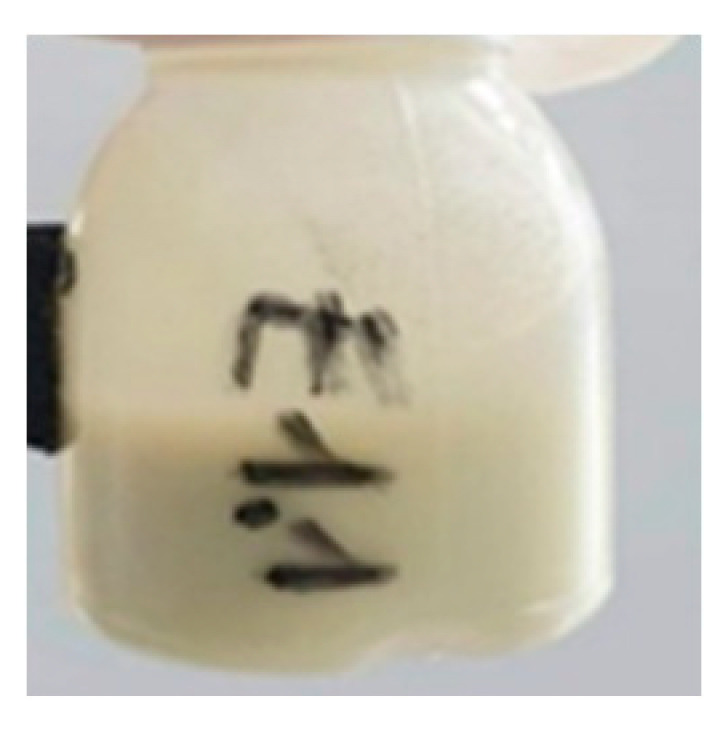	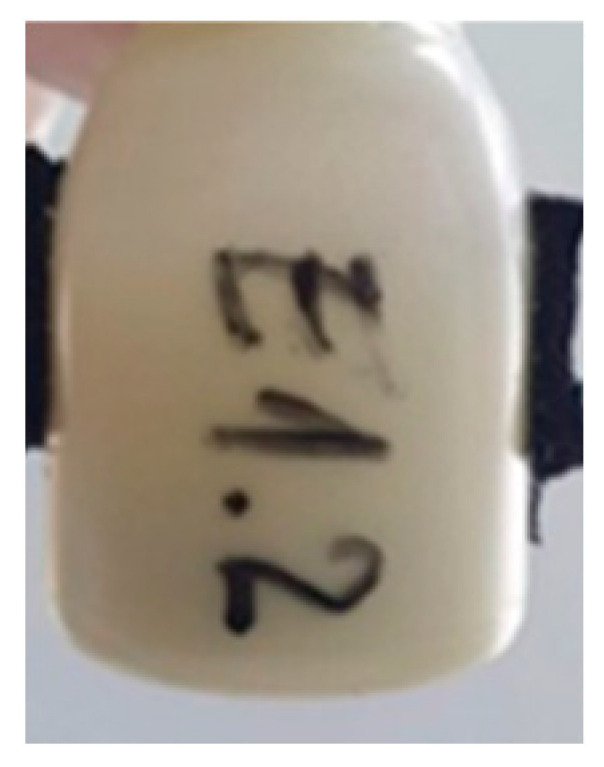	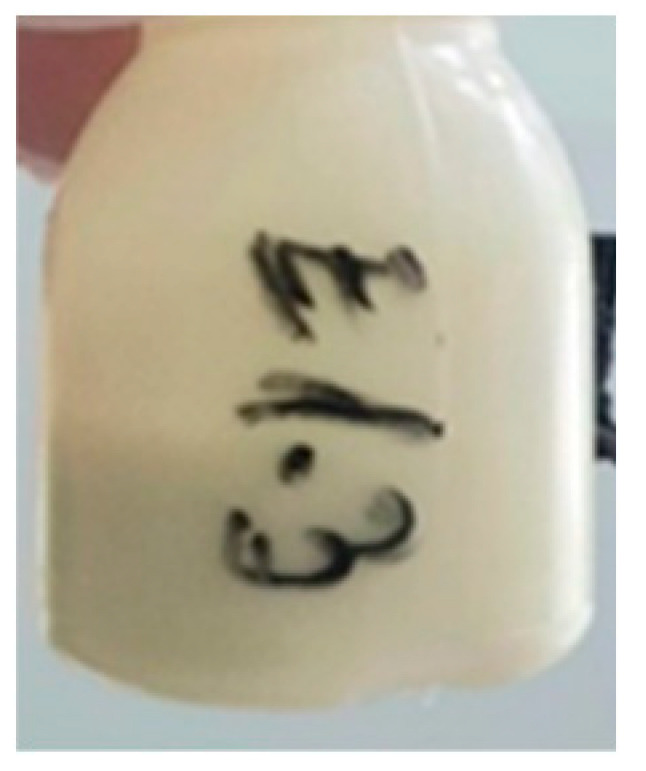

**Table 2 ijms-24-07004-t002:** TPC and antioxidant activity of dermato-cosmetic emulsions.

Sample	TPC (mg GAE/g Emulsion)	DPPH (mg TE/g Emulsion)	ABTS (mg TE/g Emulsion)
E.1.0.	0.53 ± 0.09	0.61 ± 0.03	1.72 ± 0.07
E.1.1.	6.43 ± 0.25	1.81 ± 0.01	6.12 ± 0.03
E.1.2.	10.25 ± 0.94	2.23 ± 0.03	6.50 ± 0.06
E.1.3.	11.53 ± 0.88	2.10 ± 0.04	6.31 ± 0.18

Data are presented as mean ± standard deviation (SD) of three determinations. Abbreviations: ABTS—2,2′-azino-bis(3-ethylbenzothiazoline) 6-sulfonic acid; DPPH—1,1-diphenyl-2-picrylhydrazyl; GAE, gallic acid equivalents; TE—Trolox equivalents; TPC—total phenolic content.

**Table 3 ijms-24-07004-t003:** The result of microbiological tests (plate reading after 24/48 h after inoculation) for the sample belonging to the studied dermato-cosmetic emulsions.

Sample	Total Viable Microbiological Count, CFU/g	Total Viable Bacteria Count, CFU/g	Total Viable Yeast and Mold Count, CFU/g	Presence of Pathogenic Contaminants
E 1.0.	80	80	0	absent
E 1.1.	40	40	0	absent
E 1.2.	0	0	0	absent
E 1.3.	0	0	0	absent

**Table 4 ijms-24-07004-t004:** Optical microscopy image of studied emulsions (magnification—1000×).

Dermato-Cosmetic Emulsion Samples			
E.1.0.	E.1.1.	E.1.2.	E.1.3.
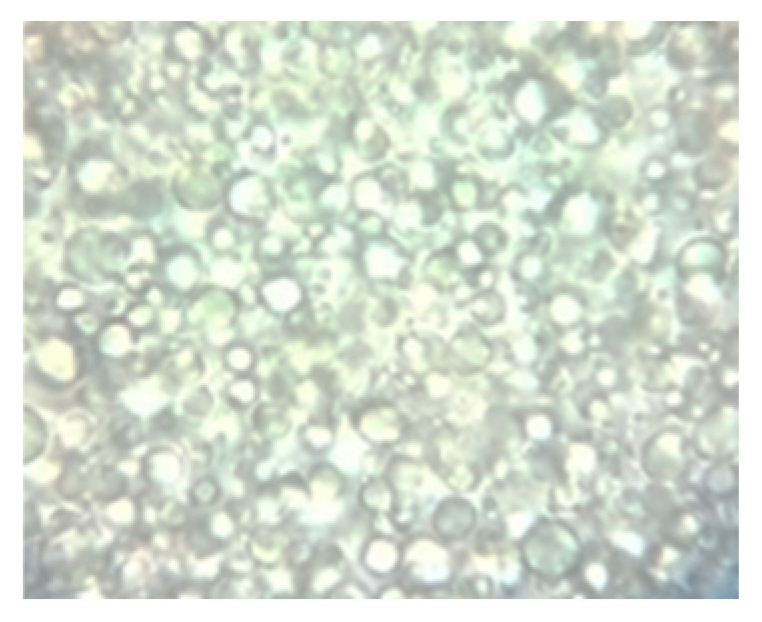	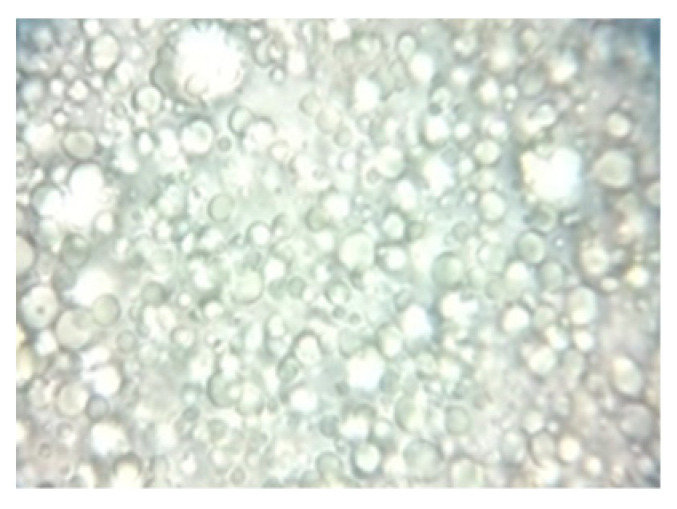	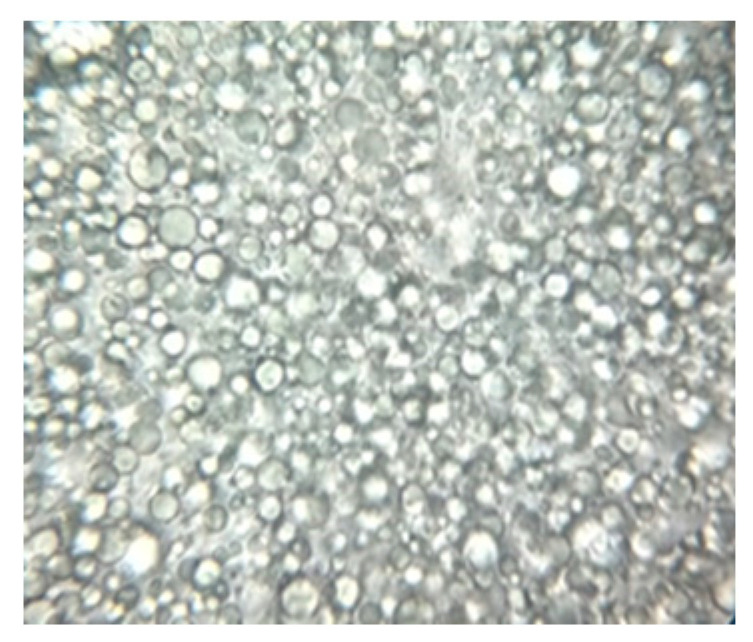	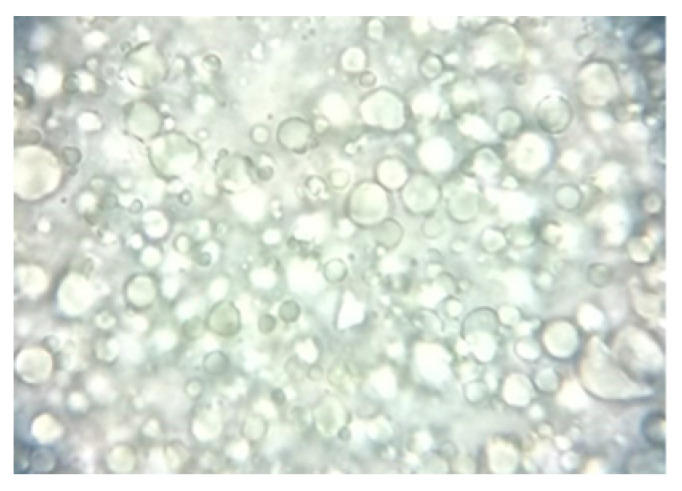

Abbreviations: E.1.0.—emulsion without active substances; E.1.1.—emulsion with 0.5% BAK and 0.5% TPA; E.1.2.—emulsion with 1% BAK+ 1% TPA; E.1.3.—emulsion with 1% BAK + 2% TPA

**Table 5 ijms-24-07004-t005:** Ingredients of O/W dermato-emulsions formulation.

% (*w*/*w*)	Function	INCI Name of Compounds
Dispersed Phase 23.0%	Emulsifiant	Polyglyceryl-6 Stearate, Polyglyceryl-6 Behenate
	Co-emulsifiant	Glyceryl stearate
	Oil phase	*Euterpe oleracea oil*
		*Oenothera biennis oil*
		*Punica granatum seed oil*
Continuous Phase 73.2%	Aqueous phase	*Rosa damascena hydrosol*
	Thickening agent	Xanthan gum
Phase C 3.8%	Active ingredients	BAK and TPA in the established concentration
	Preservative	benzyl alcohol, salicylic acid, glycerin, sorbic acid

## Data Availability

Not applicable.

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
