# Peer review of "Preliminary Approaches to Cosmeceuticals Emulsions Based on N-ProlylPalmitoyl Tripeptide-56 Acetat-Bakuchiol Complex Intended to Combat Skin Oxidative Stress"

_ijms, 2023, doi:10.3390/ijms24087004_

Round 1
Reviewer 1 Report
I recommend the manuscript titled "Preliminary approaches about cosmeceuticals emulsions based on N-ProlylPalmitoyl Tripeptide-56 Acetat - Bakuchiol complex intended to combat skin oxidative stress" to be accepted after revision done. Therefore, I propose:
1. Introduction:
- Is there a concern that dermo-cosmetics have a shorter shelf life? Or do they deteriorate faster? There is no information about the stability of these preparations and the possibility of using them after opening the cosmetics.
- page 2, lines 96-97: is there a risk of cancer when improving the potential of cosmetics with increase cell production substances?
- page 2, line 95: thanks is used in the wrong context, it should be through.
2. Methods:
- page 4, line 155 abbreviation HLB what is it? There is no explanation.
- Figure 2, tables 2, 3, 4, 5 - no explanation what is E1, E1.1, etc.
3. Results and discussion: No discussion or commentary of centrifugation test, vibration test, antioxidant activity results. Especially increase of antioxidant activity is worth a broader discussion and discussion.
Reviewer 2 Report
The manuscript systematically evaluated the emulsions of BAK-TPA complexes. It is well-organized and well-written and of interest to the journal's audiences. I would recommend accepting it after the minor corrections are addressed.
L18: "emulsions"
L150: The structure in Figure 1 has to be redrawn.
L173: "15 rpm"
L322:Please reorganize this sentence.
L363: Please rewrite the sentence "The effects will ...)
L405: " would be assured..."
L488: "The determinations..."
L499: " 1mm2" has to be corrected
L475: "TPC ug/5 mL ( )" has to be double checked
Reference session: please keep the citations' format consistent as the journal requested. Please check it throughout the whole session.
